# Propranolol Ameliorates the Antifungal Activity of Azoles in Invasive Candidiasis

**DOI:** 10.3390/pharmaceutics15041044

**Published:** 2023-03-23

**Authors:** Venkatesh Mayandi, Wen-Tyng Kang, Darren Shu Jeng Ting, Eunice Tze Leng Goh, Myoe Naing Lynn, Thet Tun Aung, Jamuna Vadivelu, Veluchamy Amutha Barathi, Anita Sook Yee Chan, Rajamani Lakshminarayanan

**Affiliations:** 1Ocular Infections and Anti-Microbials Research, Singapore Eye Research Institute, Singapore 169856, Singapore; 2Ophthalmology and Visual Sciences Academic Clinical Program, Duke-NUS Graduate Medical School, Singapore 169857, Singapore; 3Department of Medical Microbiology, Faculty of Medicine, University of Malaya, Lembah Pantai, Kuala Lumpur 50603, Malaysia; 4Birmingham and Midland Eye Centre, Birmingham B18 7QH, UK; 5Academic Unit of Ophthalmology, Institute of Inflammation and Ageing, University of Birmingham, Birmingham B15 2TT, UK; 6Translational Ophthalmic Pathology-Immunology Platform, Singapore Eye Research Institute, Singapore 169856, Singapore; 7Translational Pre-Clinical Model Platform, Singapore Eye Research Institute, Singapore 169856, Singapore; 8Singapore National Eye Centre, Singapore 169856, Singapore; 9Department of Pharmacy, National University of Singapore, Singapore 117559, Singapore

**Keywords:** antifungal therapies, synergism, propranolol–itraconazole, *Candida albicans*

## Abstract

The effectiveness of current antifungal therapies is hampered by the emergence of drug resistance strains, highlighting an urgent need for new alternatives such as adjuvant antifungal treatments. This study aims to examine the synergism between propranolol and antifungal drugs, based on the premise that propranolol is known to inhibit fungal hyphae. In vitro studies demonstrate that propranolol potentiates the antifungal activity of azoles and that the effect is more pronounced for propranolol–itraconazole combination. Using an in vivo murine systemic candidemia model, we show that propranolol–itraconazole combination treatment resulted in a lower rate of body weight loss, decreased kidney fungal bioburden and renal inflammation when compared to propranolol and azole treatment alone or untreated control. Altogether, our findings suggest that propranolol increases the efficacy of azoles against *C. albicans*, offering a new therapeutic strategy against invasive fungal infections.

## 1. Introduction

Over the past two decades, the incidence of fungal infections is growing exponentially owing to the expanding populations of immunocompromised patients, extended use of immunosuppressive agents and broad-spectrum antibiotics and increased use of invasive devices and implants [1,2]. Candidemia, a blood-stream infection caused by *Candida albicans*, is the leading cause of morbidity and mortality in health-care systems and is associated with increased costs of care and duration of hospitalization [3,4,5,6]. Currently, five major categories of antifungal drugs are available to treat invasive fungal infections [7,8,9]. Of the five categories of antifungals polyenes, azoles and allylamines target the ergosterol or ergosterol biosynthetic pathways, whereas caspofungins target the β-D-glucan synthase and 5-fluorocytosine targets the DNA [7,10,11]. However, these antifungal drugs had only moderate success in reducing high mortality rates due to invasive fungal infections and have not kept pace with the increased incidence of drug resistance. The excessive use of a limited choice of antifungals has led to the widespread evolution of multidrug-resistant fungal strains [12]. Therefore, there is an unmet clinical need to develop novel antifungal drugs or a combination of current drugs for present and future therapy.

Combination therapy is one of the most successful approaches to combat infectious diseases. It has been shown that the amphotericin B/flucytosine combination increased the rate of yeast clearance and improved survival among patients with cryptococcal meningitis [13]. A few drugs/compounds have been shown to display synergism with fluconazole when tested in combinations. For example, Gamarra et al. have shown that amiodarone (an antiarrhythmic drug) and fluconazole combination therapy showed synergism when tested against fluconazole-resistant fungal strains both in vitro and in vivo [14]. The immunosuppressant cyclosporine A was shown to be synergistic with azoles, caspofungin and amphotericin B but caused alopecia [15,16]. Zhang et al. reported the synergism between the antitumor drug geldanamycin and fluconazole against fluconazole-resistant *C. albicans* [17]. Chen et al. showed that posaconazole showed marked synergism with calcineurin inhibitor (FK506) [18]. Quan et al. reported the synergism between plant alkaloid berberine chloride and fluconazole [19]. Subsequently, Li et al. showed synergism between antimalarial drug, chloroquine and fluconazole combinations that were effective against fluconazole-resistant strains [20]. More recently, Revie et al. showed that a combination of azole and imidazopyrazoindole reverses azole resistance in *C. albicans* [21]. Previously, we showed that the addition of mitochondrion inhibitor sodium azide rigidifies the cytoplasmic membrane of *C. albicans* and conferred complete protection from membrane-targeting antifungal peptide, B4010 [22]. However, the addition of a membrane fluidizer (benzyl alcohol) reversed the protective action of NaN3 and rescued the antifungal activity of B4010. It is likely that the changes in membrane fluidity may affect the drug susceptibilities of *C. albicans*. Therefore, we hypothesize that agents that fluidize the cytoplasmic membrane of *C. albicans* may potentiate the antifungal activities of antifungals. It has been shown that β-blockers interact with the lipids and fluidize the eukaryotic model membranes [23]. In this work, we examined the potential synergism of various combinations of beta-blockers and antifungal drugs against *C. albicans*. We identified that propranolol strongly synergized with the azole antifungal drugs against *C. albicans*, with maximum synergism observed between propranolol and itraconazole. The efficacy of the propranolol–itraconazole combination was then tested in a renal abscess model, with promising efficacy in terms of reduced fungal bioburden and renal abscesses.

## 2. Materials and Methods

### 2.1. Antifungal Agents and Cultures

Sabouraud’s dextrose (SD) broth was purchased from Difco, USA. Antifungal agents (amphotericin B, natamycin, terbinafine hydrochloride, fluconazole, voriconazole and itraconazole) and propranolol were purchased from Sigma Chemical Co. (St Louis, MO, USA). Drugs were obtained in powder form, and stock solutions were adjusted to the required concentration, depending on the potency of each tested drug.

### 2.2. Antifungal Susceptibility Testing

MICs of the individual drugs were determined in Sabaroud’s dextrose broth (SDB). Susceptibility testing was performed by broth microdilution methodology in accordance with the Clinical and Laboratory Standards Institute (CLSI) M27-A3 guidelines [24]. The required strain of *Candida* sp. was cultured for 24 h, and the turbidity was adjusted using a 0.5 McFarland standard. A 150× dilution was performed using SDB. A serial dilution of the required drug solution was performed. The positive control consisting of inoculum and SDB and the negative control consisting of SDB only were prepared in duplicates. The optical density of 600 nm was monitored using an Infinite M200 monochromator microplate reader (Tecan Group Ltd., Männedorf, Switzerland) for 0 h and 24 h at 37 °C. The values reported were an average of the duplicates. Percentage growth was calculated as (A_0_ − A_min_)/(A_max_ − A_min_) × 100, where A_0_ is the observed absorbance of the sample, and A_min_ and A_max_ were obtained from the positive control curve.

#### 2.2.1. Checkerboard Assay of Combined Antifungal Activity

The checkerboard assay was performed for the drug combination study as previously described [25]. The stock solutions and serial two-fold dilutions of each drug to at least double the MIC were prepared. A total of 50 μL of SDB was distributed into each well of the microdilution plates. The first antibiotic of the combination was serially diluted along the ordinate, while the second drug was diluted along the abscissa. An inoculum equal to a 0.5 McFarland turbidity standard was prepared from each *Candida* isolate in SDB. Each microtiter well was inoculated with 100 μL of a fungal inoculum of 5 × 10^5^ CFU/mL, and the plates were incubated at 37 °C for 24 h. The growth on 0 h and 24 h were measured using a TECAN machine.

#### 2.2.2. Time-Kill Kinetics Study

Time-kill study of β-blocker (propranolol) and azoles against *C. albicans* DF2672R was tested singly and in combination, as previously described with slight modification. In brief, each drug was prepared in concentrations of 1/4× and 1/8× the MIC in reaction tubes with sterile water for injection (pH = 7.2). Each reaction tube was inoculated with 100 µL of overnight cultured *C. albicans* cells adjusted to a 0.5 McFarland standard, yielding a final cell density of ~10^3^–10^5^ CFU/mL. A control tube was prepared with only the inoculum and SDB. All tubes were incubated at 37 °C for 24 h. A 100 µL sample was withdrawn at t = 0 and 24 h from the control tube and served as the initial and final count. A 100 µL sample was withdrawn from the other tubes at various time intervals (1, 2, 4, 8 and 24 h incubation with both drugs). Samples withdrawn at each time point were subjected to a 10×, 100× and 1000× serial dilution with phosphate buffer. A 100 µL sample was withdrawn from each dilution and plated with SDA in duplicates using the pour plate method. The plates were allowed to incubate at 37 °C for 24 h. Average counts from the lowest dilution were taken and used to calculate the CFU/mL in each tube at each time point. A graph of log CFU/mL against time was then plotted.

### 2.3. Fluorescence Titration Experiments

To determine propranolol–azole interactions, a titration experiment was carried out to quantify the strength of interactions. To 500 μM of propranolol solution (200 μg/mL) in a stirred cuvette, 1 μL of concentrated stock solution of azoles was added, and the fluorescence spectra was recorded at an emission wavelength of 300–470 nm at an excitation wavelength of 290 nm by using a Quanta Master spectrofluorometer. The fluorescence spectra were recorded for each azole drug added to propranolol. The difference in fluorescence intensity was plotted against the concentration of the drug and fit into Hill’s equation to determine the binding constant.

### 2.4. Hemolytic Assay

Rabbit blood was extracted from New Zealand White rabbits. The blood was centrifuged at 3000 rpm for 10 min at 4 °C and washed twice with 20 mM PBS (phosphate-buffered saline). Respective treatment solutions were prepared 2× in the buffer and then diluted 1:1 with 8% *v/v* blood. The mixture was transferred to a 96-well microplate and incubated at 37 °C for 1 h. After incubation, the mixture was transferred to microtubes and centrifuged at 3000 rpm for 3 min to collect the supernatant. The supernatant was transferred to a new 96-well microplate and the absorbance was measured at 576 nm in the Infinite M200 monochromator microplate reader (Tecan Group Ltd., Switzerland). The percentage hemolysis was calculated using the equation:% Hemolysis = (A0 − A_min_)/(A_max_ − A_min_) × 100
where A0 is the observed absorbance, A_min_ is the average absorbance of 4% *v/v* blood without treatment and A_max_ is the average absorbance of 8% *v/v* blood diluted 1:1 with 4% Triton-X100.

### 2.5. Ethics Statement

All animals used in this study were maintained and treated in compliance with the Guide for the Care and Use of Laboratory Animals (National Research Council, Singapore) and the ARVO statement for the Use of Animals in Ophthalmic and Vision Research under the approved supervision of SingHealth Experimental Medicine Centre. The protocol was approved by the Institutional Animal Care and Use Committee (IACUC), SingHealth, Singapore (2015/SHS/1091).

### 2.6. 5-Fluorouracil Induced Immunosuppression

ICR mice were receiving 0.2 mL intravenous injection of a single dose of 5-fluorouracil (DBLTM Fluorouracil Injection BP) to render the mice neutropenic. Vials with 50 mg/mL were used without further dilutions one day prior to inoculation.

### 2.7. In Vivo Murine Model of Systemic Candidiasis

Four- to six-weeks-old ICR 20 mice (InVivos Pte Ltd., Singapore) weighing 24–40 g were used in this study. For the systemic infection model, 200 μL of *Candida albicans* ATCC 10,231 strains were cultured in SDB overnight at 30 °C and washed twice with PBS (pH 7.2), resuspended in the same buffer and the final inoculum was adjusted to 1×10^6^ CFU/mL. An amount of 200 μL of this cell suspension was injected intravenously via lateral tail-vein injection. At 4 h, 24 h, 48 h and 72 h post infection, the infected mice were treated with intraperitoneal (200 μL) once daily with the test compounds (propranolol, itraconazole and the combination) and vehicle control (0.9% saline). For dose optimization (*n* = 4 mice/group), the concentration of itraconazole (0.125, 0.25, 0.5, and 1 mg/kg) and propranolol (5, 10 and 25 mg/kg) were used. To determine the synergism, the following combination doses were used: 1.25 mg/kg propranolol and 0.0625 mg/kg itraconazole (n = 10 mice/group) as well as 0.625 mg/kg propranolol and 0.0625 mg/kg itraconazole (n = 5 mice/group). Mice conditions were observed twice daily, and the body weights of the animals were recorded every day. At day 7 post infection, all mice were euthanized with inhalational CO_2_, and the kidneys were harvested and placed in sterile 0.9% saline at 4 °C. The homogenate was then serially diluted in 1:10 concentration, and the aliquots were plated on SDA at 35 °C for 24 h for viable colony counts.

### 2.8. Grocott Methenamine Silver (GMS) Staining

Four-five µm of formalin-fixed and paraffin embedded sections were prepared for GMS special stain [26], using GMS staining kit (Ventana Medicat Systems, Inc. Arizona, United States) on Ventana BenchMark Special Stains Automated system (Roche Diagnostics Nederland BV, Almere, Netherlands). Slides were then viewed using standard light microscope (Olympus BX-40), and images were taken by Nikon Eclipse Ni-E microscope imaging system.

### 2.9. Hematoxylin and Eosin (H&E) Staining

Kidneys were harvested from mice after euthanization at day 7 post infection, the mouse kidney tissues were fixed with 10% formalin and embedded in paraffin. Kidneys were bisected vertically after fixation. The embedded tissues were then sliced into 5 μm thick sections, mounted on glass slides and incubated at 75 °C for 30 min. After deparaffinization using xylene for 10 min, the specimens were rehydrated using a graded ethanol series (95%, 85%, and 70% ethanol). After washing, the specimens were treated with hematoxylin for 2 min, washed in running tap water for 1 min and then incubated with acid alcohol for 1 s. Afterward, the specimens were incubated with ammonia water solution for 1 s and washed in running tap water for 10 min. After counterstaining in Eosin solution for 90 s, the specimens were dehydrated using 70%, 80%, 90% and 100% ethanol. Finally, the specimens were mounted with a mounting medium and observed under microscopy (10× and 100×).

### 2.10. Statistical Analysis

Statistical differences between study groups were determined using the Kruskal–Wallis multiple comparison test and Student’s *t*-test; a *p* value of 0.05 was considered to be indicative of a statistically significant result. Computations were performed using GraphPad Prism 8 (GraphPad Software, San Diego, CA, USA).

## 3. Results

### 3.1. Propranolol Is Synergistic with Azoles against C. albicans

Propranolol alone displayed a weak inhibitory activity as complete fungal inhibition was observed between 800 μg/mL whereas azole drugs exhibited trailing end points (Appendix A). To infer the synergism between propranolol and antifungal drugs, we determined the concentration of antifungals required to cause ≥90% growth inhibition in the presence of 100 μg/mL of propranolol against five different strains of *C. albicans*. At this concentration of propranolol, no or weak inhibition (≤20%) was observed against the tested strains. Table 1 reports the MIC90% values for the azole drugs in the presence of propranolol (100 μg/mL). We noted that propranolol–azole combinations resulted in complete inhibition of *C. albicans’* growth, confirming the potency of drug combinations. Polyene antifungals, however, showed no enhancement in the antifungal activity, whereas terbinafine hydrochloride displayed a moderate enhancement in the activity in the presence of propranolol (Appendix A). The synergistic interaction of propranolol with itraconazole was more pronounced when compared to voriconazole and fluconazole. To confirm this, we performed a time-kill kinetics assay. The growth rate of *C. albicans* with sub-lethal concentrations of individual drugs was determined (Figure 1a–c). A weak inhibitory activity was observed in the presence of individual drugs. In contrast, the growth decreased with the presence of propranolol and azole in combination. At concentrations of 1/4× and 1/4× MIC of propranolol–itraconazole combinations, the growth inhibition was higher than the other two azole drugs (Figure 1c). It is important to note that propranolol—itraconazole combinations achieved a reduction in viable cells (when compared to initial inoculum) even at 1/4× and 1/4× MIC combinations. The log CFU of drug combinations remained similar for 24 h of exposure at the other combinations, indicating significant inhibitory activity at these concentrations.

### 3.2. Interaction of Propranolol with Azole Antifungals

To determine if the synergistic interactions between propranolol and azole antifungals were independent or mediated by drug–drug interactions, we determined the apparent dissociation constant, Kd, by taking advantage of the intrinsic fluorescence properties of propranolol. Propranolol displayed distinct fluorescence spectra upon excitation at a wavelength of 290 nm, with three peak maxima around 328, 341 and 354 nm (Figure 2a). The fluorescence intensity of propranolol decreased gradually with increasing concentration of the azole drugs, while there was only a small change observed for the emission maximum and shape of the peaks. This suggests that propranolol could bind directly to fluconazole, voriconazole and itraconazole in a concentration-dependent manner, respectively. The Kd values of itraconazole and voriconazole were 0.13 ± 0.01 mM and 0.32 ± 0.04 mM, respectively (Figure 2b–d). Compared with both of these azole drugs, fluconazole exhibited higher Kd values, 3.6 ± 0.08 mM, confirming a weaker binding interaction with propranolol. The binding affinity of azoles further corroborates with the higher synergism observed between the propranolol and itraconazole combination.

Hemolytic activity of propranolol, antifungals and their combination. Prior to the in vivo testing, we determined the hemolytic activity of the drugs alone and their combinations for rabbit erythrocytes. Fluconazole, voriconazole and propranolol did not show any hemolytic activity even at 1 mg/mL. However, itraconazole caused 50.2 ± 0.34% and 72.0 ± 0.44% hemolytic activity at 0.5 and 1.0 mg/mL, respectively (Appendix A). Next, we determined the hemolytic activities of propranolol–antifungal combinations. The results suggested that propranolol–fluconazole was the least hemolytic, followed by propranolol–voriconazole combinations, whereas significant hemolytic activity was observed for propranolol–itraconazole combinations (Figure 3a–c). Taken together, these results suggest that a safe concentration of drugs in vivo needs to be optimized to determine the efficacy of the drug combinations.

Development of renal abscess model of *C. albicans* in mice. *C. albicans* (ATCC 10231) were administered intravenously into a lateral caudal tail vein at an inoculum density of 5 × 10^6^ CFU/mouse. Kidney fungal burden increased from 4.2 ± 0.7 log CFU/kidney at day 1 to 4.8 ± 0.6 log at day 4 and reached a value of 5.2 ± 0.5 log at day 7 post infection (Figure 4a). The gross morphology of the infected kidney appeared swollen, mottled in color and contained a number of infectious foci at day 7 p.i. (Figure 4b). When compared to naïve non-infected mice, a significant decrease in average body weight was observed in the infected mice at day 7 p.i. (Figure 4c).

To determine the optimum drug concentration, we first monitored the average body weight loss with increasing concentration of itraconazole and propranolol. Mice injected with 0.25 or 0.5 mg/kg of itraconazole intraperitoneally displayed less decrease in average weight loss when compared to untreated control and mice that were treated with higher or lower concentration of the drug (Figure 5a). In addition, all (100%) mice treated with itraconazole (0.125 mg/kg) survived until the experimental end point at day 7 compared to those treated with 0.25 mg/kg (75% survival), 0.5 mg/kg (25% survival) and 1 mg/kg (25% survival). Therefore, a lower concentration (<0.125 mg/kg) of itraconazole was chosen to be used in combination with propranolol. These results suggest a possible toxic effect of the drug at higher concentrations. Mice treated with propranolol (5, 10 and 25 mg/kg) displayed no decrease in average weight loss relative to untreated mice, suggesting a lack of antifungal activity or toxicity related to the drug (Figure 5b). In addition, after necropsy, we did not observe any adverse effects in the vital organs even after treatment with elevated concentrations of propanolol or propranolol–azole combination.

Propranolol–itraconazole combination decreases fungal bioburden in a murine systemic candidemia model. Next, we examined the efficacy of combination therapy in comparison to monotherapy. Propranolol (1.25 mg/kg) and itraconazole (0.0625 mg/kg) were used as monotherapy and in combination to determine the synergistic interactions between the two drugs. To confirm the efficacy of the combination, animals were randomized and divided into four groups including the untreated, itraconazole treated (0.0625 mg/kg), propranolol treated (1.25 mg/kg) and the combination (i.e., 0.0625 mg/kg itraconazole and 1.25 mg/kg propranolol) groups. The results demonstrated a substantial decrease in weight loss in mice that were treated with itraconazole, propranolol and combination therapy. The effect was most pronounced in the case of combination therapy (Figure 5c). It should be noted that only 30% of the mice that received combination therapy had ≥15% average weight loss, which was significantly lower than the monotherapy group (60–70%) and the control group (100%) (Table 2). In addition, combination treatment decreased the kidney fungal burden. The fungal burden for the untreated mice was 5.59 ± 0.81 log CFU/kidney, which was similar to the mice treated with propranolol (5.42 ± 0.48 log CFU/kidney) and itraconazole (5.61 ± 0.71 log CFU/kidney). The value, however, decreased to 4.74 ± 0.51 log CFU/kidney for the mice that received combination therapy. When compared to the untreated group, combination treatment resulted in an ~11.6-fold decrease in fungal burden, whereas monotherapy with propranolol or itraconazole resulted in a 3.4- or 1.2-fold decrease in the fungal burden. The gross examination of the kidneys of mice that received combination treatment showed that they appeared much healthier at the experimental end point compared to the other treatment groups, in which off-white *Candida* lesions/abscesses were clearly visible (Figure 5e and Appendix A). A combination of propranolol (0.625 mg/kg) and itraconazole (0.0625 mg/kg) did not decrease the fungal burden or improve the gross morphology of the kidney, suggesting that the previous concentration was optimum for antifungal regimens (Appendix A).

Histopathological examination of *C. albicans* infected and untreated kidneys displayed severe inflammation, presence of white *Candida* lesions, tissue damage and fungal pseudohyphae (Figure 6a–d and Appendix A). Mice kidneys that received monotherapies (Appendix A) contained much less pseudohyphae and predominantly only *C. albicans* yeast-forms, further confirming the efficacy of combination therapy (Figure 6e–l). However, mice kidneys treated with combination therapy contained reduced inflammation (Appendix A) and hardly fungal pseudohyphae and smaller clusters of *C. albicans* yeast forms than those with monotherapies, confirming the potent effects of combination antifungal activity (Figure 6m–p).

## 4. Discussion

There is an urgent need to develop alternative treatment options for fungal infections owing to a rapidly increasing rise of antifungal-resistant pathogens. In this study, we identified propranolol as an azole-potentiating agent in vitro and established its antifungal efficacy in combination with itraconazole in a mice model of a renal abscess. The rationale for the choice of propranolol was based on previous reports that it alters the membrane fluidity of phospholipid bilayers by modifying the intermolecular hydrogen bonding network and orientation of P-N dipole of phospholipid molecules as well as its ability to inhibit the formation of hypha by *C. albicans* [27,28,29]. We showed that propranolol potentiates the activity of azole antifungal drugs, while a weaker effect was observed for polyene and terbinafine hydrochloride. A recent study by Bao et al., reported a weak synergism between propranolol and natamycin against *Fusarium solani*, corroborating our results [30]. Since azole antifungals have trailing end points (often expressed as MIC50 or MIC80), we determined the potentiating activity of propranolol at a concentration that was non-lethal to *C. albicans*. The presence of 100 μg/mL of propranolol abrogated the trailing end point of the azole drugs and a >90% inhibitory effect was observed, confirming the synergistic activity of the combination. Propranolol–azoles combination may evoke numerous other responses in the cell. Time-kill kinetics study of the propranolol-azoles combination therapy showed that the growth rate of *C. albicans* decreased as compared to the individual drug alone, whereby the growth rate increased substantially after 4 h of incubation. Among the three azole antifungals, the effect was more pronounced for the propranolol–itraconazole combination, followed by voriconazole and fluconazole. Fluorometric titrations provide information on the interaction of the drug membranes or drug–drug combinations, as shown in many studies previously [28,31,32]. This study confirmed that azole antifungal drugs interact with propranolol with varying affinities. The decreased fluorescence intensity observed in this study supported the binding interaction between propranolol and azoles. Itraconazole and voriconazole displayed greater binding affinity than fluconazole. Thus, the enhanced activity observed for the propranolol–itraconazole combination is attributed to the higher affinity associated with the two drug combinations than fluconazole and voriconazole. The alteration in the physical state of liposomal membrane lipids is one of the factors that seem to affect the azole susceptibility of *C. albicans* [31].

Systemic candidemia model in rodents indicated substantial weight loss, major changes in the gross morphology of kidneys and increased fungal burden in the untreated mice. Spellberg et al. have shown that mice infected with invasive *C. albicans* SC5314 strains resulted in a ≥15% weight loss at the time of death [33]. Therefore, we included ≥15% weight loss as one of the key pathophysiological parameters of the disease. Progressive ≥15% weight loss occurred in mice that were untreated/saline-treated, whereas 60% or 70% of mice treated with propranolol or itraconazole alone resulted in weight loss, respectively. However, only 30% of mice treated with a combination of drugs showed a weight loss of ≥15%, suggesting a better therapeutic effect. The kidney fungal burden reached the maximum at 7 days post infection for the untreated mice and was unaltered by treatment with propranolol or fluconazole monotherapies. Combination treatment resulted in a significant decrease in renal fungal burden compared to untreated mice, confirming the efficacy of the treatment.

Histological examination revealed the presence of fungal pseudohyphae in kidney tissues of untreated mice with a pronounced inflammatory response. This effect was markedly reduced in the presence of propanol–itraconazole treated mice with evidence of infection resolution. Hyphae formation is one of the major virulence traits of *Candida* sp. [34]. Propanolol has been shown to interact with acidic phospholipids with greater affinity than neutral phospholipids [32]. The interaction of propranolol with phosphatidic acid, a key lipid associated with the dimorphic transition in *C. albicans*, inhibits the formation of hyphal without altering the growth rate. Interestingly, a decrease in MIC was observed for azoles and terbinafine hydrochloride, which impair the sterol biosynthesis, and not with the antifungals that bind to the membrane sterol. These observations suggest that propranolol may potentiate the activity of azoles by modifying the ergosterol biosynthetic pathways, a mechanism that was similar to imidazopyrazoindole NPD827 [21].

## 5. Conclusions

We showed that the presence of propranolol potentiates the antifungal activity of azole drugs against *C. albicans*. A strong interaction between propranolol and itraconazole may be responsible for the enhanced antifungal activity. In vitro and in vivo experiments demonstrate that the combination treatment decreased the vital pathophysiological parameters such as higher weight loss, kidney fungal burden and inflammation of vital organs. This could potentially pave the way for a new treatment strategy for treating systemic fungal infections. Future research on molecular mechanisms and detailed in vivo studies may advance our understanding of the role of β-blockers–azole combinations, and thus enable us in discovering new antifungal treatments with the promise of combating drug resistance while remaining non-toxic to humans.

## Figures and Tables

**Figure 1 pharmaceutics-15-01044-f001:**
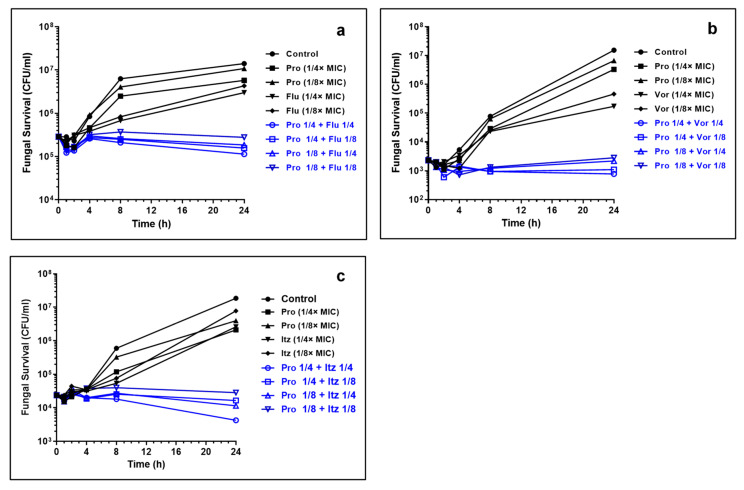
Time-kill kinetics of propranolol–azole combinations. (**a**) Propranolol (Pro) and fluconazole (Flu); (**b**) propranolol and voriconazole (Vor) and (**c**) propranolol and itraconazole (Itz). Note substantial lethality observed at sub-lethal concentrations of Pro–Itz combinations. The concentrations of propranolol, itraconazole and voriconazole are expressed in terms of their MIC values.

**Figure 2 pharmaceutics-15-01044-f002:**
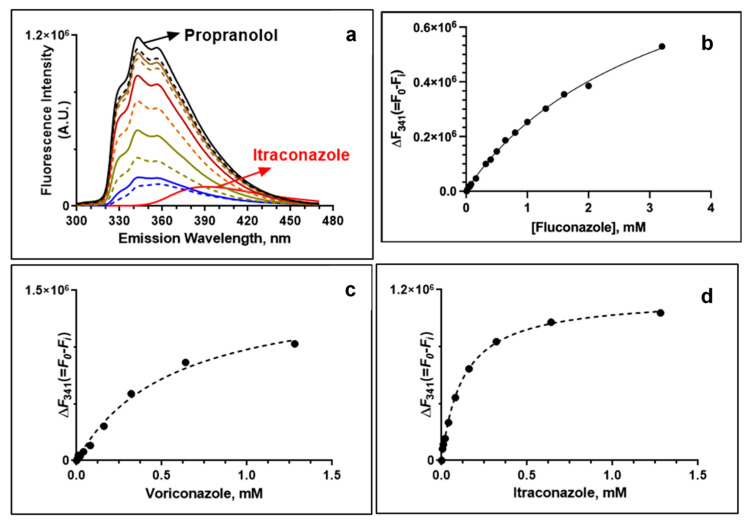
Interactions between propranolol and azole drugs monitored by fluorescence spectroscopy. (**a**) Fluorescence scan of propranolol upon addition of increasing concentrations of itraconazole. Fluorescence titration of azole drugs showing the change in fluorescence intensity (at 341 nm) of propranolol: (**b**) propranolol and fluconazole; (**c**) propranolol and voriconazole (Vor) and (**d**) propranolol and itraconazole (Itz).

**Figure 3 pharmaceutics-15-01044-f003:**
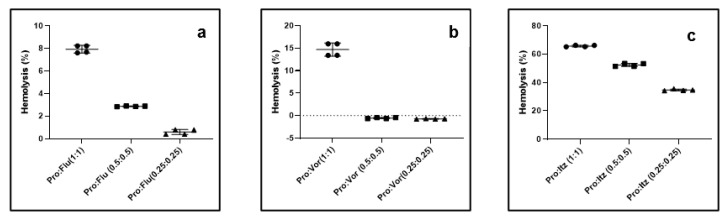
Hemolytic activity of propranolol–azole antifungal combinations for rabbit erythrocytes. (**a**) Propranolol–fluconazole; (**b**) propranolol–voriconazole and (**c**) propranolol–itraconazole. The value indicates the drug concentration in mg/mL.

**Figure 4 pharmaceutics-15-01044-f004:**
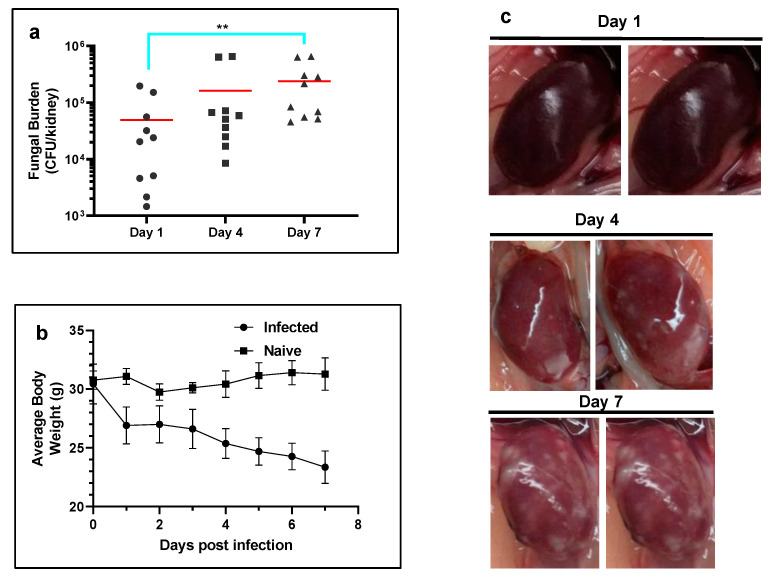
Development of systemic candidemia model in mice. (**a**) Temporal change in the kidney fungal burden of mice (n = 5 mice per group) infected with *C. albicans*. The red horizontal bar indicates the mean value. Note the significant increase in fungal burden at day 7 p.i. (** *p* ≤ 0.01 by Kruskal–Wallis multiple comparison test). (**b**) Average body weights of mice infected with *C. albicans* (n = 5 per group). (**c**) Temporal change in the gross morphology of kidneys infected with *C. albicans.*

**Figure 5 pharmaceutics-15-01044-f005:**
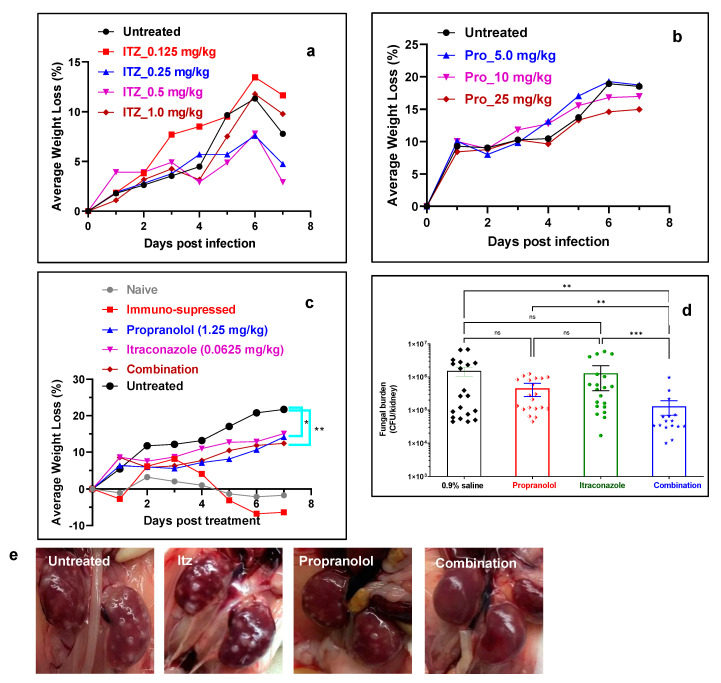
Efficacy of antifungal mono and combination therapies. (**a**) Average body weight loss (%) of infected mice after treatment with Itz monotherapy (n = 4 mice per group). (**b**) Average body weight loss (%) of infected mice after treatment with propranolol monotherapy (n = 4 mice per group). (**c**) Average body weight loss (%) of infected mice after treatment with Itz–propranolol combination monotherapy (n = 10 mice per group). (**d**) Fungal burden in kidneys after treatment with various groups (n = 20 kidneys per group). The error bar indicates mean ± SE.* *p* ≤ 0.05, ** *p* ≤ 0.01 and *** *p* ≤ 0.001 by Kruskal–Wallis multiple comparison test. (**e**) Photographs of kidneys after treatment with various groups.

**Figure 6 pharmaceutics-15-01044-f006:**
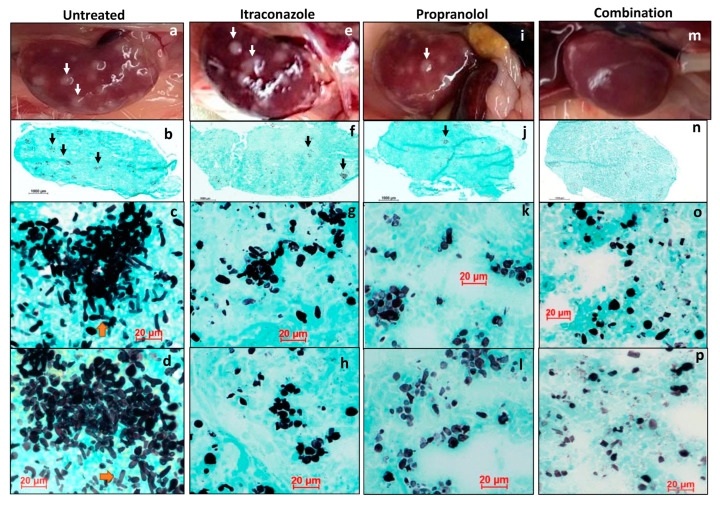
Grocott-Gomori’s methenamine silver (GMS) stain for fungi load. Untreated kidney (**a**–**d**): (**a**) gross image of the kidney with white infective foci (white arrow); (**b**) whole mount of kidney, Grocott-Gomori’s methenamine silver (GMS) stain for fungi shows small colonies of fungi (black arrows); (**c**,**d**) representative images, GMS stain, 40× magnification, high power view of the large collections of fungal elements composed of numerous pseudo-hyphae and elongated yeast cells. Itraconazole treated kidney (**e**–**h**): (**e**) gross image of the kidney with white infective foci (white arrow); (**f**) whole mount of kidney, GMS stain for fungi shows fewer colonies of fungi (black arrows); (**g**,**h**) representative images, GMS stain, 40× magnification, high power view of the smaller fungal clusters that show fewer pseudo-hyphae and a larger proportion of elongated yeast cells. Propranolol treated kidney (**i**–**l**): (**i**) gross image of the kidney with white infective foci (white arrow); (**j**) whole mount of kidney, GMS stain for fungi shows fewer colonies of fungi (black arrows); (**k**,**l**) representative images, GMS stain, 40× magnification, high power view of the smaller fungal clusters show occasional pseudo-hyphae and a predominance of elongated yeast cells. Combination treated kidney (**m**–**p**): (**m**) gross image of the kidney with white infective foci (white arrow); (**n**) whole mount of kidney, GMS stain for fungi shows fewer colonies of fungi (black arrows); (**n**–**p**) representative images, GMS stain, 40× magnification, high power view of the fungal elements show occasional clusters of fungal elements composed mainly of elongated yeast cells. In comparison with itraconazole- and propranolol-treated kidneys (**e**–**l**), there is a lower fungal load in combination treatment (**m**–**p**).

**Table 1 pharmaceutics-15-01044-t001:** Summary of the synergism between propranolol and azole antifungals. MIC ≥ 90% (μg/mL) of azoles in the presence of 100 μg/mL propranolol.

Fungal Strains	Fluconazole	Voriconazole	Itraconazole
*C. albicans* 1976R	12.5	3.125	0.78
*C. albicans* 2672R	12.5	3.125	0.78
*C. albicans* ATCC 2091	6.25	12.5	0.78
*C. albicans* ATCC 24433	12.5	1.56	0.78
*C. albicans* ATCC 10231	12.5	3.125	0.78

**Table 2 pharmaceutics-15-01044-t002:** Body weight loss of mice untreated or treated with antifungals.

Group	Number of Mice with an Average Weight Loss of ≥15% at Day 7 p.i. (n = 10 per Group)
Untreated	10
Propranolol	6
Itraconazole	7
Propranolol + Itraconazole	3

## Data Availability

Not applicable.

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
