# Peer review of "Propranolol Ameliorates the Antifungal Activity of Azoles in Invasive Candidiasis"

_pharmaceutics, 2023, doi:10.3390/pharmaceutics15041044_

Round 1

Reviewer 1 Report

This manuscript, number Pharmaceutics-2251339, entitled: “Repurposing Propranolol and Azole Combination for Invasive Candidiasis” by Mayandi et al., showed with in vitro and in vivo studies that propanolol potentiate the antifungal activity of azoles and its effect is more pronounced due to propanolol-itraconazole combination. The authors suggest this combination as a new therapeutic strategy against invasive fungal infection by C. albicans.

Here are the review points

Major comments:

1) In the Materials and Methods section (2.1 Antifungal susceptibility testing), What broth microdilution method was used? For example: “The MICs were determined by the microbroth dilution technique as described by CLSI M27-A3”. Please include this information because standardization of the inoculum is critical for MIC’s assays.

2) For MIC’s assays, the typical medium used is RPMI-1640. Why was  the use of this culture medium not considered?

3) Why did use an inoculum of 5 x 105 CFU/mL? According to the CLSI M27-A3,  for the MICs must be 0.5 x103 – 2.5 x103. The inoculum and culture medium used are critical for this type of assay. Why was the reference method for this type of essay not included?

Clinical and Laboratory Standards Institute (CLSI). Reference method for broth dilution antifungal susceptibility testing of yeasts. Approved standard-third edition. Document M27-A3. Wayne, PA: CLSI; 2008.

4) In Figure 1 of the Results section: if all the essays have the same initial inoculum, why are there variations in the fungal survival of the controls at time 0? Wouldn't it be better to use the propanol-voriconazole combination?

Minor comments:

Page 2, line 83, change parentheses order in the following sentence “Antifungal agents (amphotericin B, natamycin, terbinafine hydrochloride, fluconazole, voriconazole and itraconazole and propranolol)” for “Antifungal agents (amphotericin B, natamycin, terbinafine hydrochloride, fluconazole, voriconazole, and itraconazole) and propranolol”

Page 2, line 89, change “Candida” for “Candida sp.”

Page 2, line 89, what culture medium is it? "SDB"

Page 3, line 106, change “bacterial inoculum” for “fungal inoculum”

Page 3, line 139, change “10 mins” for “10 min”

Page 3, line 142, change “1 hour” for “1 h”

Page 4, line 182, change “35 C” for “35 °C”

Page 14, line 537, change “Candida ” for “Candida spp.”

Author Response

Response to Reviewer 1 Comments

Manuscript Title:

Propranolol Ameliorates the Antifungal Activity of Azoles in an Invasive Candidiasis Manuscript Number: Pharmaceutics-2251339

On behalf of all authors, I would like to thank you all for giving your time in evaluating our manuscript and providing valuable comments which really helps in improving the quality of the manuscript. We would like to thank the editor for an opportunity to address the queries raised by the reviewers. We have addressed all the comments raised by the reviewers and highlighted in the revised manuscript. A point-by-point response to the reviewers’ comments is given below. Please note that the changes have been made in blue colour in the revised manuscript.

Response to Reviewer 1

Comments and Suggestions for Authors

This manuscript, number Pharmaceutics-2251339, entitled: “Repurposing Propranolol and Azole Combination for Invasive Candidiasis” by Mayandi et al., showed with in vitro and in vivo studies that propanolol potentiate the antifungal activity of azoles and its effect is more pronounced due to propanolol-itraconazole combination. The authors suggest this combination as a new therapeutic strategy against invasive fungal infection by C. albicans.

Here are the review points

Major comments:

Comment 1: In the Materials and Methods section (2.1 Antifungal susceptibility testing), What broth microdilution method was used? For example: “The MICs were determined by the microbroth dilution technique as described by CLSI M27-A3”. Please include this information because standardization of the inoculum is critical for MIC’s assays.

Response: Thank you very much for pointing this out. We have done the changes as per the reviewer’s suggestion. (Line# 88-90)

Comment 2: For MIC’s assays, the typical medium used is RPMI-1640. Why was the use of this culture medium not considered?

Response: We thank the reviewer for the comments. However, as per the CLSI protocol for Candida albicans Sabaroud’s dextrose broth is recommended over RPMI-1640 medium. In addition, it has been shown that the growth of planktonic growth of C. albicans was maximum in Sabaroud’s dextrose broth. (Mem Inst Oswaldo Cruz. 2016 Nov; 111(11): 697–702. Published online 2016 Oct 3. doi: 10.1590/0074-02760160294).

Comment 3: Why did use an inoculum of 5 x 105? According to the CLSI M27-A3, for the MICs must be 0.5 x103 – 2.5 x103. The inoculum and culture medium used are critical for this type of assay. Why was the reference method for this type of essay not included?

Clinical and Laboratory Standards Institute (CLSI). Reference method for broth dilution antifungal susceptibility testing of yeasts. Approved standard-third edition. Document M27-A3. Wayne, PA: CLSI; 2008.

Response: An inoculum size of 5 × 105 was used so as to determine the antifungal activity in a high density of inoculum. Our results are further supported by the in vivo experiments that the average kidney burden ranged from 4.5×104 to 6.8×106 (n=20) after tail vein injection.

We thank the reviewer for pointing out this. As suggested by the reviewer, we have now referenced the article in the revised manuscript. (Line #656-657)

Comment 4: In Figure 1 of the Results section: if all the essays have the same initial inoculum, why are there variations in the fungal survival of the controls at time 0? Wouldn't it be better to use the propanol-voriconazole combination?

Response: We concur with the reviewer’s comment. As the experiments were performed in different days, we could observe some variations in the inoculum size. We thank the reviewer for pointing this error. We have revised this in the manuscript. (Line #118)

Minor comments:

Comment 5: Page 2, line 83, change parentheses order in the following sentence “Antifungal agents (amphotericin B, natamycin, terbinafine hydrochloride, fluconazole, voriconazole and itraconazole and propranolol)” for “Antifungal agents (amphotericin B, natamycin, terbinafine hydrochloride, fluconazole, voriconazole, and itraconazole) and propranolol”

Response: Thank you very much for pointing this error. We have modified the text, as per the reviewer’s suggestion. (Line #83)

Comment 6: Page 2, line 89, change “Candida” for “Candida sp.”

Response: Thank you very much for pointing this out. We have modified the text, as per the reviewer’s suggestion. (Line #91)

Comment 7: Page 2, line 89, what culture medium is it? "SDB"

Response: Thank you very much for pointing this out. We have modified the text, as per the reviewer’s suggestion. (Line #88)

Comment 8: Page 3, line 106, change “bacterial inoculum” for “fungal inoculum” Page 3, line 139, change “10 mins” for “10 min”

Response: Thank you very much for pointing this out. We have modified the text, as per the reviewer’s suggestion. (Line #108)

Comment 9: Page 3, line 142, change “1 hour” for “1 h”

Response: Thank you very much for pointing this out. We have modified the text, as per the reviewer’s suggestion. (Line #144)

Comment 10: Page 4, line 182, change “35 C” for “35 °C”

Response: Thank you very much for pointing this out. We have modified the text, as per the reviewer’s suggestion. (Line #185)

Comment 11: Page 14, line 537, change “Candida ” for “Candida spp.”

Response: Thank you very much for pointing this out. We have modified the text, as per the reviewer’s suggestion. (Line #536)

Reviewer 2 Report

Dear authors,

please check the following information:

In section 2.2 of materials and methods (Antifungal susceptibility testing), Candida should be written in italics.

In section 2.2.1. Checkerboard assay of combined antifungal activity, line 106, bacterial inoculum, in this case it would be fungal inoculum?

On line 174 of section 2.7. In vivo murine model of systemic candidiasis, specify which was the vehicle control.

  In Table 1 it is important to add the MIC ≥90% of Propranolol

In the legend of figure 1 it is not clear what the authors mean when they add ¼ and 1/8 to the text, please specify this.

Candida on lines 480, 537 568 and 570 must be in italics

Despite the authors having carried out the hemolytic effect, it would be very important to carry out a cytotoxicity assay in Vero cells or murine fibroblasts, the cytotoxic effect of Propanolol is not very clear.

Put species names in italics in references

Author Response

Response to Reviewer 2 Comments

Manuscript Title:

Propranolol Ameliorates the Antifungal Activity of Azoles in an Invasive Candidiasis Manuscript Number: Pharmaceutics-2251339

On behalf of all authors, I would like to thank you all for giving your time in evaluating our manuscript and providing valuable comments which really helps in improving the quality of the manuscript. We would like to thank the editor for an opportunity to address the queries raised by the reviewers. We have addressed all the comments raised by the reviewers and highlighted in the revised manuscript. A point-by-point response to the reviewers’ comments is given below. Please note that the changes have been made in blue colour in the revised manuscript.

Response to Reviewer 2

Dear authors,

please check the following information:

Comment 1: In section 2.2 of materials and methods (Antifungal susceptibility testing), Candida should be written in italics.

Response: Thank you very much for pointing this out. We have modified the text, as per the reviewer’s suggestion. (Line #91)

Comment 2: In section 2.2.1. Checkerboard assay of combined antifungal activity, line 106, bacterial inoculum, in this case it would be fungal inoculum?

Response: Thank you very much for pointing this out. We have modified the text, as per the reviewer’s suggestion. (Line #108)

Comment 3: On line 174 of section 2.7. In vivo murine model of systemic candidiasis, specify which was the vehicle control.

Response: Thank you very much for pointing this out. We have modified the text, as per the reviewer’s suggestion. (Line #176)

Comment 4: In Table 1 it is important to add the MIC ≥90% of Propranolol

Response: We thank the reviewer for pointing this. We observed a complete inhibition of C. albicans at a high concentration of propranolol an it was difficult to quantify MIC99 for propranolol.

Comment 5: In the legend of figure 1 it is not clear what the authors mean when they add 1⁄4 and 1/8 to the text, please specify this.

Response: We thank the reviewer pointing this. The fractions 1/4 and 1/8 are indicated as 1/4th or 1/8th of their respective MIC values. The figure caption is corrected in the revised manuscript. (Line #264-265)

Comment 6: Candida on lines 480, 537 568 and 570 must be in italics

Response: Thank you very much for pointing this out. We have modified the text, as per the reviewer’s suggestion. (Line #479, 536, 567 and 569)

Comment 7: Despite the authors having carried out the hemolytic effect, it would be very important to carry out a cytotoxicity assay in Vero cells or murine fibroblasts, the cytotoxic effect of Propanolol is not very clear.

Response: We thank the reviewer for pointing this. Propanolol alone did not show any haemolytic activity even at elevated concentration (1 mg/mL). This has been indicated in the text. (Line #303)

As we do not have the access to Vero cells and murine fibroblasts, we did not carry out the tests in these cells.

Comment 8: Put species names in italics in references

Response: Thank you very much for pointing this out. We have done the changes as per the reviewer’s suggestion. (Line #632, 634, 635, 638, 640, 643, 646, 665, 668, 673 and 680).

Reviewer 3 Report

The manuscript “Repurposing Propranolol and Azole Combination for Invasive Candidiasis” is an interesting article because it examines the synergy between propranolol and antifungal drugs (azoles), based on the history that propranolol inhibits fungal hyphae. The results obtained in this work are encouraging from a therapeutic point of view since it is a useful strategy to counteract resistance to antifungal drugs. It is a well-written paper, which analyzes the synergism between propranolol and azoles against C. albicans and the interaction of propranolol and azoles. However, I do have an observation: the authors should discuss the effects that propranolol, a drug commonly used for high blood pressure, could cause in healthy individuals since its administration in them has not been tested.

I suggest changing the job title:

Synergism between propranolol and azole combination for invasive candidiasis

Minor comments:

Italicize Candida albicans, C. albicans in the text and in references.

Line 401: Change (c) to (c)

Line 458: Change Figure 6 to Figure 6.

Author Response

Response to Reviewer 3 Comments

Manuscript Title:

Propranolol Ameliorates the Antifungal Activity of Azoles in an Invasive Candidiasis Manuscript Number: Pharmaceutics-2251339

On behalf of all authors, I would like to thank you all for giving your time in evaluating our manuscript and providing valuable comments which really helps in improving the quality of the manuscript. We would like to thank the editor for an opportunity to address the queries raised by the reviewers. We have addressed all the comments raised by the reviewers and highlighted in the revised manuscript. A point-by-point response to the reviewers’ comments is given below. Please note that the changes have been made in blue colour in the revised manuscript.

Response to Reviewer 3

Comment 1: The manuscript “Repurposing Propranolol and Azole Combination for Invasive Candidiasis” is an interesting article because it examines the synergy between propranolol and antifungal drugs (azoles), based on the history that propranolol inhibits fungal hyphae. The results obtained in this work are encouraging from a therapeutic point of view since it is a useful strategy to counteract resistance to antifungal drugs. It is a well-written paper, which analyzes the synergism between propranolol and azoles against C. albicans and the interaction of propranolol and azoles. However, I do have an observation: the authors should discuss the effects that propranolol, a drug commonly used for high blood pressure, could cause in healthy individuals since its administration in them has not been tested.

Response: We concur with the reviewer’s comment. After necropsy, we did not observe any adverse effect in vital organs of the animals treated with propranolol alone or propranolol:azole combination. This has been added in the text. (Line #369-371)

Comment 2: I suggest changing the job title:

Synergism between propranolol and azole combination for invasive candidiasis

Response: We thank the reviewer for the comments. However, we did not see a clear synergism in our in vitro studies. Therefore, we have changed the title as “Propranolol ameliorates the antifungal activity of azoles in an invasive candidiasis.” (Line #2-3)

Minor comments:

Comment 3: Italicize Candida albicans, C. albicans in the text and in references.

Response: Thank you very much for pointing this out. We have done the changes as per the reviewer’s suggestion. (Line #632, 634, 635, 638, 640, 643, 646, 665, 668, 673 and 680).

Comment 4: Line 401: Change (c) to (c)

Response: Thank you very much for pointing this out. We have done the changes as per the reviewer’s suggestion. (Line #400)

Comment 5: Line 458: Change Figure 6 to Figure 6.

Response: Thank you very much for pointing this out. We have done the changes as per the reviewer’s suggestion. (Line #457)
